# Spectroscopic Ellipsometry Studies on Solution-Processed OLED Devices: Optical Properties and Interfacial Layers

**DOI:** 10.3390/ma15249077

**Published:** 2022-12-19

**Authors:** Maria Gioti

**Affiliations:** Nanotechnology Laboratory LTFN, Physics Department, Aristotle University of Thessaloniki, GR-54124 Thessaloniki, Greece; mgiot@physics.auth.gr; Tel.: +30-2310-998103

**Keywords:** F8:F8BT, PFN-Br, OLED, ellipsometry, slot-die, blend, interfacial properties

## Abstract

Τhe fabrication of organic light-emitting diodes (OLEDs) from solution involves the major problem of stack integrity, setting the determination of the composition and the characteristics of the resulting interfaces prerequisite for the optimization of the growth processes and the achievement of high devices’ performance. In this work, a poly(9,9-dioctylfluorene) (F8) and poly(9,9-dioctylfluorene-alt-benzothiadiazole) (F8BT) blend is used for the emissive layer (EML), poly-3,4-ethylene dioxythiophene; poly-styrene sulfonate (PEDOT:PSS) is used for a hole transport layer (HTL), and Poly(9,9-bis(3′-(N,N-dimethyl)-N-ethylammoinium-propyl-2,7-fluorene)-alt-2,7-(9,9-dioctylfluore-ne))dibromide (PFN-Br) for an electron transport layer (ETL) to produce the OLED device. All the layers are developed using the slot-die process, onto indium tin oxide (ITO)-coated polyethylene terephthalate (PET) flexible substrates, whereas Ag cathode was formed by ink-jet printing under ambient conditions. Spectroscopic ellipsometry measurements were performed upon completion of the successive films’ growth, in sequential steps, for the multilayer OLED development. Ellipsometry analysis using different models demonstrate the degree of intermixing within the layers and provide information about the interfaces. These interfacial properties are correlated with the emission characteristics as well as the final performance of the OLED devices.

## 1. Introduction

In recent years, conjugated polymers have attracted high interest among many scientists in the field of organic semiconductors. The characteristic features of these polymers are environmental stability, solution processability, easy solubility, and potential for use on flexible substrates. Their unique and tailored optoelectronic properties make them suitable to use in solid-state lighting devices such as organic light-emitting diodes (OLEDs) [1,2,3]. Increasing demands for energy saving and for new and efficient energy sources towards green lighting make successful, large volume-production OLEDs, by solution-based roll-to-roll processes, a high priority [4]. 

OLED devices contain many layers of different functionality apart from electrodes, such as a hole transport layer (HTL), emissive layer (EML), and electron transport layer (ETL). However, the fabrication of these multilayers from solution involves the major problem of stack integrity, since the sequential layers are soluble in common organic solvents and/or chemical interdiffusion may occur, leading to the significant interfacial widths. By alternating the solvent used (e.g., water and organic solvent) for each layer, we are able to prevent the redissolution of the layer in the next processing step and finally the sequential layers can be successfully processed. However, there is always a possibility of partial intermixing between the layers after the fabrication of the stack, instead of the formation of a multilayer with distinct interfaces [5]. As a consequence, multilayer architectures with well-defined interfaces or with controllable interfacial thicknesses are the key challenge for the solution deposition process. Thus, the composition and the characteristics of the resulting interfaces of solution-processed layers are critical factors in determining device performance and achieving high efficiencies. In addition, further advances in the design and fabrication of OLED structures require the accurate knowledge of the optical properties of all materials used in the sequential layers. Prior to the design of device architecture for its fabrication, it is important to elucidate the underlying nature of solution-processed thin films either as a layer or as an intermixing layer that is formed during the growth of the multilayer structures. 

Spectroscopic ellipsometry (SE) is one of the research methods that can be used to determine all these crucial issues as it is an especially useful, fast, non-destructive measurement technique for studying the optical properties and thickness of samples and enabling the evaluation of the morphology of the samples [6,7,8]. By applying the proper modelling and fitting analysis recipes, we can achieve a comprehensive characterization of interfacial layers [9]. This is of high importance, as the whole approach could be easily implemented in situ and in real-time, providing the control of roll-to-roll processes and successfully enhancing mass production. It should be noted here that there are peculiar problems with the ellipsometric data analyses employed while studying polymer films grown on transparent substrates, due to the low optical contrast, which follows from the approximate equality of the refractive indices of the two optical media involved. In these cases, a more detailed acquisition of experimental data is required [10,11] and thus the application of in situ and real-time recording and analysis becomes unfeasible. 

In this paper, the SE characterization of fully solution processed OLED device with a blended EML of poly(9,9-di-n- -2,7-diyl) (PFO or F8) and poly(9,9-dioctylfluorene-alt-benzothiadiazole) (F8BT) is presented. For HTL poly-3,4-ethylene dioxythiophene, poly-styrene sulfonate (PEDOT:PSS) was used and as for ETL, poly(9,9-bis(3′-(N,N-dimethyl)-N-ethylammoinium-propyl-2,7-fluorene)-alt-2,7-(9,9-dioctylfluorene)) dibromide (PFN-Br) was used. These polymer films ensure that valuable insights on the interfacial properties can be obtained if the appropriate model is applied. On the one hand, the films are not transparent in the whole experimental energy range; on the other hand, the refractive indices of the involved optical media ITO and PEDOT:PSS, PEDOT:PSS and F8:F8BT, F8:F8BT and PFN-Br, that form the interfaces under study exhibit satisfactory optical contrast. Thus, the comparative SE results, by applying different structural and optical models, confirm the feasibility of using SE in the solution-processed method of the OLED fabrication.

## 2. Materials and Methods

### 2.1. Inks Preparation

For the HTL, PEDOT:PSS was purchased from Heraeus (Hanau, Germany) with the commercial product name CLEVIOS PVP Al 4083. A solution of PEDOT:PSS AI 4083 mixed with ethanol in the ratio of 3:2 was prepared. For the EML, F8 and F8BT were obtained from Ossila, UK. The F8 and the F8BT were dissolved in toluene at 19:1 giving a concentration of 1.5% (*w*/*w*). For the ETL, PFN-Br was obtained from Ossila, Sheffield, UK. A mixture of polar solvents, consisting of ethanol, methanol, and isopropanol, was used to aid in film uniformity due to the different evaporation point of each solvent. The main reason for this approach is the demand of the relatively small thickness of the ETL, at the order of 15 nm. To make such a thin film, a low concentration of 0.25% *w*/*w* was chosen. Finally, a solution of silver nanoparticles was used for the printing of the cathode electrode. The solvents were purchased from Sigma–Aldrich and used as received without further purification (St. Louis, MO, USA). 

### 2.2. OLED Fabrication 

A mini roller coater from FOM Technologies (slot-die head with 13 mm shim width) was used to coat the PEDOT:PSS, F8:F8BT, and PFN-Br layers (EMLs) onto a PET/ITO flexible substrate/anode base under ambient conditions (Humidity 30–40% RH). Subsequently, the Ag cathode was formed by ink-jet printing. The structure of the fabricated OLED devices is shown in Figure 1. Arrows denote the critical interfaces, studied in this work using spectroscopic ellipsometry.

### 2.3. Characterization Techniques 

The films were characterized in terms of their optical properties using a phase-modulated spectroscopic ellipsometer by Horiba/Jobin Yvon (UVISEL, Europe Research Center—Palaiseau, France). SE spectra were acquired at an angle of incidence of 70°, in the spectral range from 1.5 to 6.5 eV with a step of 20 meV. For the emission properties evaluation, electroluminescence (EL) measurements were performed using the Hamamatsu external quantum efficiency system (C9920-12), which measures brightness and light distribution of the devices.

### 2.4. SE Modelling 

SE is a non-destructive and surface-sensitive technique useful for measure the dielectric function (ε(E)= ε_1_(E) + iε_2_(E)) of materials. SE uses polarized light to illuminate a sample and measures the changes in the polarization state for p and s polarization components of the electric field after interaction with the sample. These polarization changes are represented by the ratio *ρ* between the Fresnel reflection coefficients for p and s polarized light (*r_p_*, *r_s_*) given by the relation [12,13]:(1)$$\rho =\frac{r_{p}}{r_{s}}=\tan\Psi e^{i\Delta }$$
where Ψ represents the amplitude ratio and Δ the phase difference for the *s* and *p* polarization types before and after reflection. 

A standard application of ellipsometry is the determination of the optical constants of a material either in the form of bulk sample or a thin film. For thin films of single or multilayer structures, the measured quantity by SE is the pseudodielectric function <ε(E)>, which also accounts the effect of the films’ thickness. By applying the appropriate modeling procedures, even the layers’ thicknesses with sub-nanometer resolution in a sandwich multilayer structure can be calculated, together with the spectral dependence of the dielectric function. 

The dielectric function of the HTL PEDOT:PSS was parameterized using the dispersion equation consisting by a Drude oscillator and two Lorentz oscillators for the description of intraband (free electrons) and interband (bounded electrons) electronic transitions [14]:(2)$$\varepsilon (E)=\varepsilon_{1}(\infty )+\frac{E_{p}^{2}}{-E^{2}+i\Gamma_{D}E}+\sum_{j=1}^{2}\frac{f_{j}E_{0j}^{2}}{E_{0j}^{2}-E^{2}+i\gamma_{j}E}$$
where *ε*_1_(∞) is the background dielectric constant; *E_p_* is the plasma energy; *Γ_D_* is the coefficient of the Drude oscillator, related to energy relaxation process; *E*_0*j*_, *f_j_* and *γ_j_* are the coefficients of the *j*th Lorentz oscillator, related to resonance photon energy, oscillation intensity, and its energy width, respectively. 

The dielectric function of EML F8:F8BT and ETL PFN-Br was parameterized using the appropriate number of the modified Tauc–Lorentz (TL) oscillator model [12,15] with energy-depended broadening [7,16]: (3)$$\varepsilon_{2}(E)=\frac{AE_{0}^{}\Gamma {\left(E-E_{g}\right)}^{2}}{{\left(E^{2}-E_{0}^{2}\right)}^{2}+\Gamma_{}^{2}E^{2}}\cdot \frac{1}{E},\;E>E_{g}$$
(4)$$\varepsilon_{2}(E)=0,\;E\leq E_{g}$$
(5)$$\Gamma \equiv \Gamma_{j}^{’}\left(E\right)=\Gamma_{j}\exp\left(-\frac{1}{1+\sigma^{2}}{\left(\frac{E-E_{j}}{\Gamma_{j}}\right)}^{2}\right)$$

The real part of the TL dielectric function is obtained by Kramers–Kronig integration (Equation (6)) and its analytical expression is given [15]:(6)$$\varepsilon_{1}(E)=\varepsilon_{1}\left(\infty \right)+\frac{2}{\pi }P\int_{E_{g}}^{\infty }\frac{\xi \varepsilon_{2}(\xi )}{\xi^{2}-E^{2}}d\xi$$

The characteristic parameters of the TL dispersion model are the energy position of the fundamental gap *E_g_*, the resonance energy *E*_0_, the broadening Γ, and the strength *A* of the oscillator, which describes the electronic transition. *ε*_1_(∞) accounts for the contribution of all electronic transitions which take place at higher energies, above the experimentally measured energy range and are not considered in the theoretically calculated *ε*_2_(*E*). σ varies from zero to infinity and more specifically for σ = 0, a Gaussian line shape is derived, whereas for σ > 5, a Lorentzian broadening is established [16,17].

Ellipsometric measurements were conducted first on the ITO-coated PET, which is the reference for the substrate, and subsequently after every layer deposition during the building of the OLED stack. Figure 2 illustrates the successive development of the multilayer OLED structure. The SE analysis was focused mainly on the study of the interfaces PEDOT:PSS/F8:F8BT and F8:F8BT/PFN-Br. For this purpose, three different geometrical and structural models were used to investigate the formation of interfacial layers. 

The first model (Model#1) refers to the identical flat and sharp interfaces. The second one (Model#2) refers to the formation of a mixed interface between the overlying and underlying layer, which is a composite film, and it is considered as an equivalent homogeneous layer with an effective dielectric function *ε_eff_*(*E*) derived by an adequate effective medium model, such as the Bruggeman Effective Medium Approximation (BEMA) [18]. According to BEMA, the effective dielectric function of a composite layer consisting of two phases is given by the equation [19]: (7)$$f_{a}\frac{\varepsilon_{a}-\varepsilon_{eff}}{\varepsilon_{a}+2\varepsilon_{eff}}+f_{b}\frac{\varepsilon_{b}-\varepsilon_{eff}}{\varepsilon_{b}+2\varepsilon_{eff}}=0$$
where *ε_a_* and *ε_b_* are the dielectric functions of the constituent materials *a* and *b* with the respective volume fractions *f_a_* and *f_b_*. 

The third model (Model#3) corresponds to a graded-layer model with a lateral variation of the composition, and the analysis of the <*ε*(*E*)> spectrum is realized using BEMA in combination to a linear gradient model. In this model the volume fractions of the two constituents have been approximated to change linearly from the bottom to the top of the interfacial layer, which consists of 10 sub-layers (or slices) [20,21].

All the experimental spectra were fitted using DeltaPsi 2.6.8 software which comes with experimental instrumentation, and by applying the Levenberg–Marquardt minimization algorithm. The quantity used to describe the agreement between the experimental data (exp) and the modeling (th) process is defined as the unbiased estimator *x*^2^ of the measured <*ε*(*ω*)>_exp_ through the expression: (8)$$x^{2}=\frac{1}{N-p-1}\sum_{i=1}^{n}\left[{\left(<\varepsilon_{1}{>}_{th}-<\varepsilon_{1}{>}_{\exp}\right)}_{i}^{2}+{\left(<\varepsilon_{2}{>}_{th}-<\varepsilon_{2}{>}_{\exp}\right)}_{i}^{2}\right]$$
where *N* is the number of the experimental points or spectral points, and *p* is the number of energy-independent (unknown) parameters describing better the whole system. The smaller standard deviation of the optimization results *x*^2^ was the criterion of the suitability and reliability of each model.

## 3. Results and Discussion

As has already been stated, SE measurements were conducted first on the ITO-coated PET and subsequently after the formation of the PEDOT:PSS layer. Figure 3a shows the ellipsometric data and the fitted theoretical ones based on the dispersion equation given by Equation (3). By this analysis, the bulk dielectric function, shown in Figure 3b, and the thickness of the PEDOT:PSS (52.3 ± 0.8 nm), were obtained. 

Subsequently, the F8:F8BT of the 19:1 ratio thin layer was deposited on PET/ITO/PEDOT:PSS and the respective SE measurements were conducted. Figure 4 (a) shows the ellipsometric data and the fitted ones. The coincidence between the experimental and the fitted spectra is almost identical, and absolutely the same for both models. This is verified by the fitting deviations between the *<ε(E)> experimental* and the <*ε(E)> fit* data using the Model#1, assuming a sharp interface, and Model#2, which accounts for the existence of an interlayer between the PEDOT:PSS and F8:F8BT layers, which are presented in Figure 4b. Figure 4c depicts the geometrical structures used for the fitting analysis and the derived numerical results. Consequently, in the case of a possible formation of a mixed interlayer between the PEDOT:PPS and F8:F8BT layers, the thickness of this ultra-thin interfacial layer is 2.0 ± 0.3 nm, with a composition of 36% F8:F8BT and 64% PEDOT:PSS, and the thicknesses of the F8:F8BT and PEDOT:PSS layers are modified to 23.4 ± 0.2 nm and to 50.9 ± 0.2 nm, from 24.1 ± 0.2 nm and 52.3 ± 0.8 nm, respectively, which were calculated by using Model#1 with the sharp interface. It is noteworthy that the x^2^ values, calculated through the fitting procedures using Model#1 and Model#2, are more or less the same. Thus, it can be concluded that no significant interdiffusion and modifications occur onto PEDOT:PSS upon the formation of the overlying F8:F8BT layer. Obviously, the use of Model#3 cannot contribute to gain further insights concerning the intermixing of the layers in this stage of the OLED structure construction. Finally, in Figure 4d is plotted the bulk dielectric function of the 19:1 F8:F8BT, calculated using the best-fit results of the parameters of the dispersion equation consisted of 5 modified TL oscillators [6], which was applied for the parameterization of its dielectric response.

Continuing with the construction of the multilayer OLED structure, the PFN-Br ETL is grown onto F8:F8BT EML. The new collected SE data were analyzed firstly using the respective geometrical model assuming the formation of a sharp interface (Model#1) and secondly by introducing an interfacial mixed layer (Model#2) between the F8:F8BT and PFN-Br layers. The derived results are presented in Figure 5. More specifically, Model#2 leads to a better fitting of the experimental data, as demonstrated by the calculated deviations between experimental and best-fit regenerated <*ε*(*E*)> spectra (see Figure 5a), which are illustrated in Figure 5b and the corresponding *x*^2^ values presented in Figure 5c. The PFN-Br film thickness is calculated to be 9.5 ± 0.3 nm by Model#1, whereas from Model#2 we obtain the formation of an interlayer 31.7 ± 2.1 nm thick, having a composition of 70% F8:F8BT and 30%PFN-Br. The underlying F8:F8BT and overlying PFN-Br layers are 1.7 ± 1.0 nm and 1.0 ± 0.5 nm thick, respectively. These results indicate an extensive intermixing between the EML and ETL. We, therefore, set to investigate the formation of a linear gradient layer by employing Model#3 for the fitting of the SE data. Based on this model, we obtain the minimization of the deviations between the experimental and the fitted spectra (see Figure 5b). The content of PFN-Br in the mixed layer, with a thickness of 34.6 ± 0.1 nm, changes linearly from 99.5% (0.5% F8:F8BT) at the top to 2.7% (97.3% F8:F8BT) at the bottom. Figure 5d depicts the calculated dielectric function spectrum of the PFN-Br using the best-fit results of the parameters of the dispersion equation, consisting of 2 modified TL oscillators, that was applied for the parameterization of its dielectric response.

The fully printed OLED device was characterized in terms of its emission profile and luminance and the respective results are shown in Figure 6a,b. The results were compared to that derived by the characterization of a reference OLED device that was developed using the optimized conditions for the thermal evaporation of the ETL Ca film and the cathode Ag film [6]. The emission profiles of the two OLED devices are almost the same. However, the operational characteristics, such as luminance and turn-on voltage (the bias voltage for 1 cd/m^2^) between the two devices exhibit significant differences. The maximum luminance for the fully printed OLED device is 230 cd/m^2^, whereas for the reference device it is 4025 cd/m^2^. Additionally, the turn-on voltages of the devices differ by 1.4 V, with that of the fully printed being higher at 5.1 V. It should be noted here that the thickness of the EMLs of both devices are almost the same, 30–35 nm. Thus, the main reason for the remarkable lower luminance of the fully printed one could be the non-successful formation of the PFN-Br film, due to its gradual interdiffusion to the F8:F8BT layer, as was revealed by the SE analysis. As a result, the required ultra-thin PFN-Br layer for the efficient carrier transport does not exist and the final device performance is relatively low. However, the device exhibits uniform surface emission, as is demonstrated by the inset photo, which was taken during the device operation with 12 V bias voltage. Consequently, we can speculate that the low luminance intensity can be further enhanced by increasing the PFN-Br thickness to achieve more efficient carrier transport, as it seems that the interdiffusion of the PFN-Br to the active layer does not affect the emission characteristics of the OLED device, and by assuming that the interdifussion will be saturated, leading to the successful formation of the electron transport layer.

## 4. Conclusions

The SE technique allows us to obtain effective information about the optical and electronic properties of materials by applying the appropriate dispersion equation. For the case of an organic material, the modified Tauc–Lorentz oscillator with the energy-dependent broadening proved to be the most appropriate for the precise description and interpretation of the dielectric response. In addition, based on adequate modeling, the results from the analysis of the dielectric function spectra measured for multilayer samples can be converted into knowledge about the material’s nanostructure or interfacial properties. These are of high importance because they determine the OLED devices’ operational characteristics and performance. As it is a non-destructive and non-contact technique, SE can be implemented in situ, so it is a candidate of choice for in-lab characterization, as well as for the control and the optimization of industrial processes in real-time.

## Figures and Tables

**Figure 1 materials-15-09077-f001:**
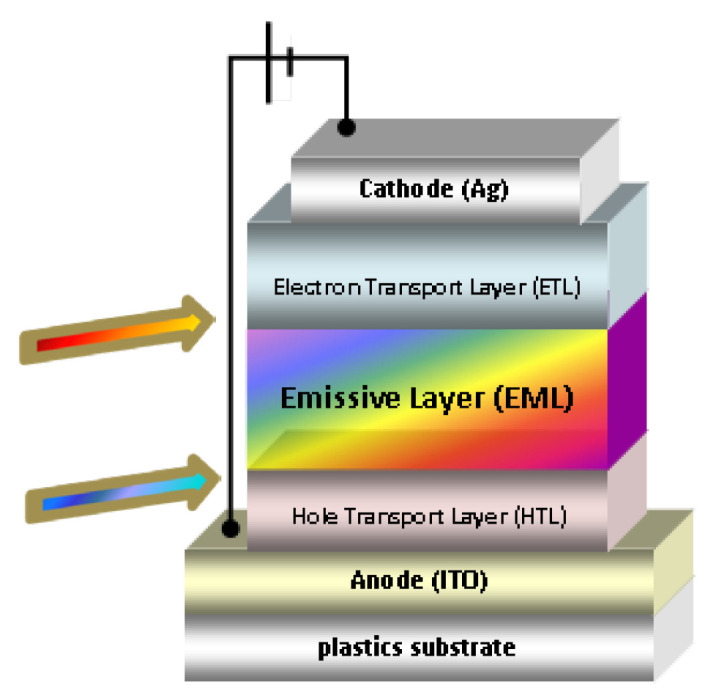
The multilayer OLED structure, with solution-processed HTL, EML, and ETL layers.

**Figure 2 materials-15-09077-f002:**
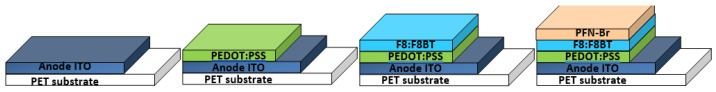
The geometrical structure of the samples for which SE measurements were performed in the sequential steps of the multilayer OLED development.

**Figure 3 materials-15-09077-f003:**
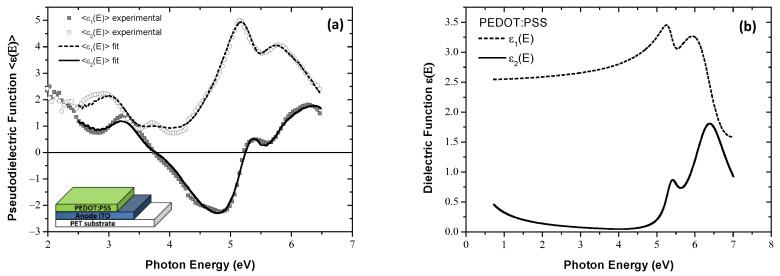
(**a**) The measured SE spectra of the PEDOT:PSS film, grown on PET/ITO (symbols) and the corresponding fitted ones (lines). Inset shows the geometrical structure of the sample. (**b**) The dielectric function ε(E) of the slot-die PEDOT:PSS, calculated using the best-fit parameters derived by SE analysis.

**Figure 4 materials-15-09077-f004:**
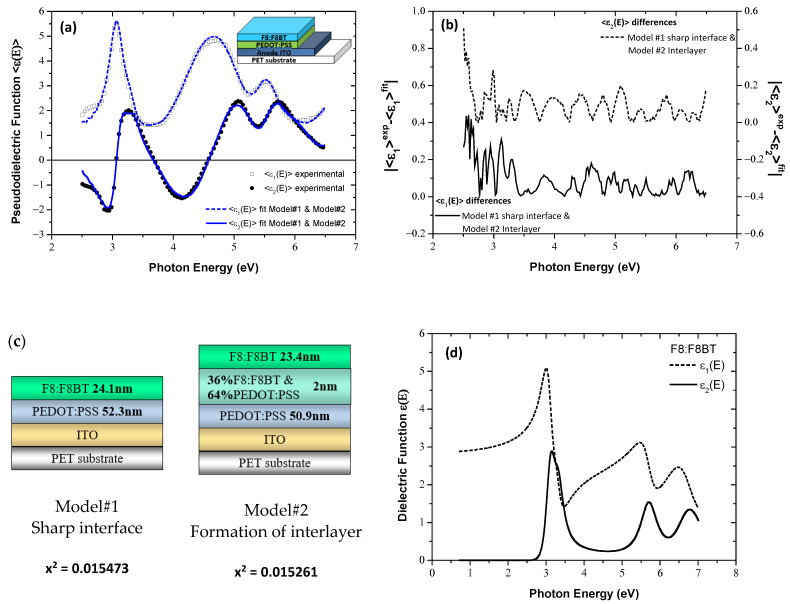
(**a**) The measured SE spectra of the F8:F8BT film, grown on PET/ITO/PEDOT:PSS (symbols) and the corresponding fitted ones (lines). Inset shows the geometrical structure of the sample. (**b**) The calculated deviations between the experimental and fitted SE spectra. (**c**) The schematical representation of the models applied in SE fitting analysis. (**d**) The dielectric function ε(E) of the slot-die F8:F8BT, calculated using the best-fit parameters derived by SE analysis.

**Figure 5 materials-15-09077-f005:**
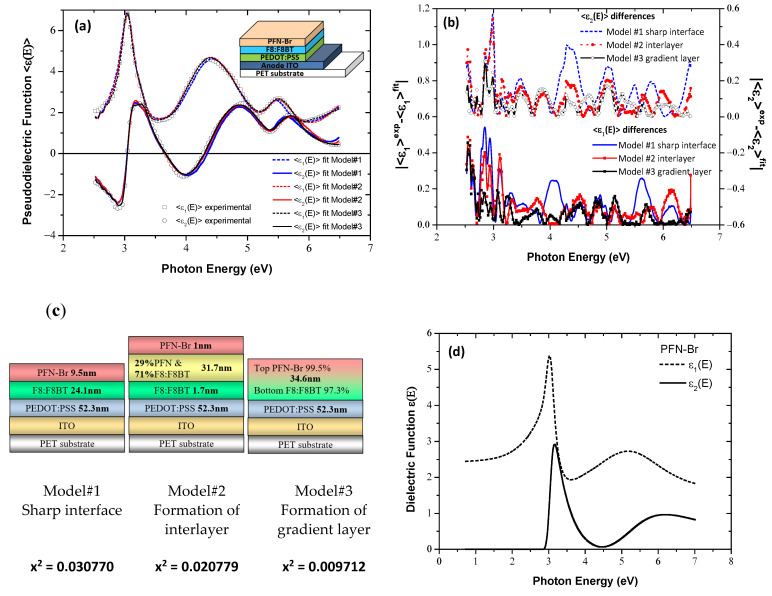
(**a**) The measured SE spectra of the PFN-Br film, grown on PET/ITO/PEDOT:PSS/F8:F8BT (symbols) and the corresponding fitted ones (lines). Inset shows the geometrical structure of the sample. (**b**) The calculated deviations between the experimental and fitted SE spectra. (**c**) The schematical representation of the models applied in SE fitting analysis. (**d**) The dielectric function ε(E) of the slot-die PFN-Br, calculated using the best-fit parameters derived by SE analysis.

**Figure 6 materials-15-09077-f006:**
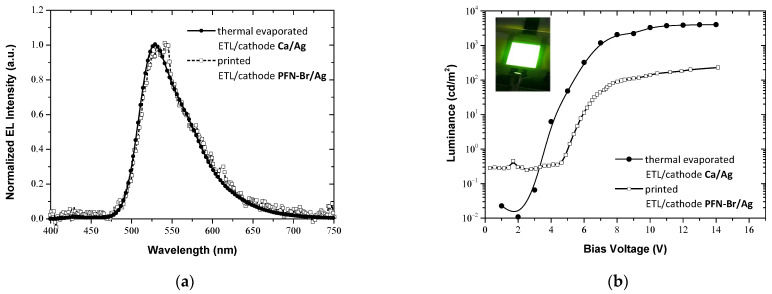
(**a**) The normalized intensities of EL emissions measured from fully printed and reference F8:F8BT OLEDs. (**b**) Luminance-voltage (L-V) plots of the fabricated fully printed and reference F8:F8BT OLEDs.

## Data Availability

Not applicable.

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
