# Peer review of "Spectroscopic Ellipsometry Studies on Solution-Processed OLED Devices: Optical Properties and Interfacial Layers"

_materials, 2022, doi:10.3390/ma15249077_

Round 1
Reviewer 1 Report
The paper titled "Spectroscopic ellipsometry studies on solution processed OLED devices: optical properties and interfacial layers" present interesting results and can be accepted for publication after major revision. Some issues should be clarified.
The first remark concerns the figure 4 expediency of showing the fitting results in Figure 4 for both models 1 and 2 since they are practically identical. The corresponding fitting curves merge with each other. Therefore, it would be better to note such a result in the text, and in the figure to show one curve as the result of the fitting procedure for both models. It is also not clear why the fitting result within model #3 for F8:F8BT film was not shown?
In the case of studying the parameters of the ETL PFN-Br film, the author is obviously dealing with the so-called ultra-thin layer. The total thickness of the structure increased by approximately 10 nm due to its coating with an ETL film. In order to design an OLED structure with a high quantum yield efficiency, it is necessary to provide even smaller thicknesses of the main layers. The phenomenon of interdiffusion between layers is obviously undesirable. Therefore, it is necessary to take into account the effect of the correlation between optical parameters and the thickness of the ultrathin layer in such ellipsometric studies. In this regard, it would be appropriate to mention a number of works in which the authors made effective attempts to overcome the correlation of the parameters of ultrathin polymer layers:
Kostruba, Andriy, Yuriy Stetsyshyn, and Rostyslav Vlokh. "Method for determination of the parameters of transparent ultrathin films deposited on transparent substrates under conditions of low optical contrast." Applied optics 54.20 (2015): 6208-6216.
Kostruba, A., Stetsyshyn, Y., Mayevska, S., Yakovlev, M., Vankevych, P., Nastishin, Y., & Kravets, V. (2018). Composition, thickness and properties of grafted copolymer brush coatings determined by ellipsometry: calculation and prediction. Soft matter, 14(6), 1016-1025
Hilfiker, James N., et al. "Determining thickness and refractive index from free-standing ultra-thin polymer films with spectroscopic ellipsometry." Applied Surface Science 421 (2017): 508-512.
Reviewer 2 Report
In this work, the authors proposed a spectroscopic ellipsometry studies on solution processed OLED devices to characterize the interlayer properties between HTL, EML and ETL. SE method was a non-destructive and non-contact technique. The authors gave the examples on how to used SE to characterize the thickness and dielectric function of different organic layers. Before further consideration, I still have several concerns as listed below:
1. The abstract should be re-organized. It is difficult to directly get the significant results from the present version.
2. Did the authors first use SE technique to characterize the OLED device?
3. How did the results from SE techniques further guide or optimize the design and preparation of high-performance OLEDs?
4. Another experiments should be performed to demonstrate the accuray of SE techniques, such as the cross sectional SEM.
5. The novelty should be further highlighted to help the readers understand the scientific significance behind this work.
Round 2
Reviewer 1 Report
The authors have answered all of my comments and the paper can be accepted for publication.
Reviewer 2 Report
The authors have mostly addressed my concerns. I recommend its publication at present form.